# RoMQA: A Benchmark for Robust, Multi-evidence, Multi-answer Question Answering

**Victor Zhong**[1,2] **Weijia Shi**[1,2] **Scott Wen-tau Yih**[2] **Luke Zettlemoyer**[1,2]

[1] University of Washington

[2] Meta AI

{vzhong, swj0419, lsz}@cs.washington.edu    scottyih@meta.com

## Abstract

We introduce RoMQA, the first benchmark for robust, multi-evidence, multi-answer question answering (QA). RoMQA contains clusters of questions that are derived from related constraints mined from the Wikidata knowledge graph. RoMQA evaluates robustness of QA models to varying constraints by measuring worst-case performance within each question cluster. Compared to prior QA datasets, RoMQA has more human-written questions that require reasoning over more evidence text and have, on average, many more correct answers. In addition, human annotators rate RoMQA questions as more natural or likely to be asked by people. We evaluate state-of-the-art large language models in zero-shot, few-shot, and fine-tuning settings, and find that RoMQA is challenging: zero-shot and few-shot models perform similarly to naive baselines, while supervised retrieval methods perform well below gold evidence upper bounds. Moreover, existing models are not robust to variations in question constraints, but can be made more robust by tuning on clusters of related questions. Our results show that RoMQA is a challenging benchmark for large language models, and provides a quantifiable test to build more robust QA methods.

## 1 Introduction

A high quality compositional question answering (QA) model should exhibit robustness to subtle variations in the underlying meaning of input questions. For exmaple, consider the question "which pianists born in Paris play Western classical music?" To show robust understanding, a QA model should not only be able to correctly answer this direct question, but also a wide range of related queries that differ in only a few constraints (e.g. who was a pianist born in Paris?, who was a Western classical pianist, not born in Paris?). Prior compositional QA datasets do not evaluate the robustness of QA models to variations in question constraints.

We introduce RoMQA, a benchmark for **Ro**bust, **M**ulti-evidence, multi-answer **QA**, that explicitly evaluates for robustness to small question perturbations. RoMQA, shown in Figure 1, differs from previous work in a number of ways.

**Evaluates robustness to constraint variations.** RoMQA contains clusters of related questions that are used to measure robustness to varying implicit question constraints. For each cluster, we compute a robustness score that is the minimum score over the questions it contains. In order to perform well on RoMQA robustness evaluation, a model must be able to understand many different combinations of the implicit constraints that define the cluster, such as what it means to be a pianist, to be born in Paris, and to play Western classical music. To our knowledge, RoMQA is the first QA benchmark that evaluates this type of robustness.

**More complex questions.** Natural questions often have many answers and cannot be answered from a single text. When compared to existing datasets, RoMQA questions have more answers (mean 108.6, median 11, as shown in Figure 2), cover a wider range of diverse topics (as depicted in Figure 3), and necessitate a higher amount of evidence text (mean 41.6, median 24). RoMQA also contains entity-linked, relation-extracted text that provide provenance for constraints, showing that the questions are answerable with multi-evidence reasoning from the text corpus.

**More natural human written questions.** Contrary to previous multi-answer compositional QA datasets, RoMQA provides a significantly larger pool of 28k human-written questions. This represents a ten-fold increase when compared to the previous leading dataset QAMParI (Amouyal et al., 2022)), which features a mere 2k human-written questions. Human evaluations show that questions in RoMQA are more natural, as gauged by

| Question | Implicit constraints | Example evidence | Answers |
|---|---|---|---|
| Which pianists born in Paris play Western classical music? | `+ occupation pianist subj`
`+ born_in Paris subj`
`+ genre western_classical subj` | Lily Maisky (born July 28, 1987 in Paris) is a classical pianist.
Anne Queffélec (born 17 January 1948) is a French classical pianist, born in Paris.
… | Lily Musky
Anne Queffélec |
| Who was a pianist born in Paris? | `+ occupation pianist subj`
`+ born_in Paris subj` | Claude Helffer (June 18, 1922 – October 27, 2004) was a French pianist noted particularly for his advocacy of 20th-century music…Helffer was born in Paris, and began piano lessons at the age of five and from the age of ten until the outbreak of World War II he studied with Robert Casadesus…
… | Gilbert Amy
Claude Helffer
… |
| Who was a Western classical mucic pianist, not born in Paris? | `+ genre western_classical subj`
`+ occupation pianist subj`
`- born_in Paris subj` | David Fray (born 24 May 1981) is a French classical pianist…David Fray was born in Tarbes, near the Pyrenees.
André Watts (born June 20, 1946) is an American classical pianist and professor at the Jacobs School of Music of Indiana University…Born in Nuremberg, Germany, Watts is the son of a Hungarian mother…
… | David Fray
André Watts
… |

Figure 1: A cluster of related questions, implicit constraints, evidence text, and answers from RoMQA. Within a RoMQA cluster, related questions differ in implicit constraints. In addition to evaluating model performance across questions, RoMQA evaluates robustness to variations in question constraints by scoring worst-case performance among related questions.

how likely a person is to ask the question (Figure 4). Qualitatively, RoMQA questions contain fewer overly precise constraints, unusual attribute comparisons, and overly large numbers of referential hops.

We evaluate state-of-the-art large language models (LMs) on RoMQA in zero-shot prompting, few-shot in-context learning, and supervised learning settings. In the closed setting where the model selects among 100 candidate entities, zero-shot and few-shot LMs perform on par (e.g. 38.5 F1 by 8-shot OPT-175B, Zhang et al. (2022)) with simple baselines such as predicting all candidate entities (33.5 F1). RoMQA also remains very challenging for state-of-the-art supervised methods, with the best retrieve-then-classify model achieving 63.8 F1 compared to a gold-evidence upper bound of 95.0 F1. The open setting, where no candidates are given, is even more challenging to existing methods — the state-of-the-art Instruct-GPT3 (`text-davinci-002`, Ouyang et al. (2022)) obtains 12.6 Pr@10 (precision at 10) while supervised retrieve-then-generate obtains 58.6 Pr@10.

Finally, no test model is robust to variations in question constraints. The best performing retrieval method obtains a worse-case related question test score of 37.9 F1 in the closed setting — a 25.9 F1 absolute drop compared to evaluating questions independently. Training on clusters of related questions, such as RoMQA clusters, improves model robustness over training on unrelated questions. However, the robustness gap remains large — closing this gap will likely require significant advances in natural language understanding. We open-source RoMQA at anonymous.url.

## 2 RoMQA

We describe RoMQA construction and how it differs from prior compositional QA datasets.

### 2.1 Dataset construction

RoMQA construction has three goals. First, we want a diverse selection of question topics. Second, these questions should require reasoning over multiple pieces of evidence. Third, we need to understand what implicit constraints the questions contain in order to evaluate robustness to varying constraints. At a high level, RoMQA construction involves 1) sampling constraints from knowledge base (KB) triples, 2) clustering related constraints, 3) sampling implicit constraints that form logical queries, and 4) annotating language questions.

**Sampling constraints from a KB.** We create RoMQA questions from Wikidata (Vrandečić and Krötzsch, 2014) that are answerable given entity-linked and relation-extracted text (Elsahar et al., 2018). Wikidata consists of subject-proposition-object triples such as `Gilbert_Amy occupation pianist`. We convert these triples into entity-relation **constraints**. For instance, the previous example is decomposed into constraints `Gilbert_Amy occupation obj` and `pianist occupation subj`.

**Clustering related constraints.** A **cluster** of related constraints shares at least two answer entities. For example, `occupation pianist subj` and `place_of_birth Paris subj` are in the same cluster because they share the same answers `Gilbert_Amy` and `Claude_Helffer` (Paris-born pianists). As Wikidata has a skewed proposition distribution, we resample cluster constraints with probability inversely proportional to their propo-

| **RoMQA** |
|---|
| A film composed by S. Thaman and produced by Ganesh Babu. |
| Who did not play for the Carolina Panthers but was a linebacker and was on the Chicago Bears? |
| Which members of the Royal Society received the Order of Australia, but were not employed by the University of Oxford? |
| Sub-orbital spaceflight that launched from Cape Canaveral Air Force Station Launch Complex 5. Launched by Mercury-Redstone Launch Vehicle |
| Who is an athlete who participated in diving, and was born in Stockholm? |

| **HotpotQA** |
|---|
| Are Random House Tower and 888 7th Avenue both used for real estate? |
| Which American singer and songwriter has a mezzo-soprano vocal range, Tim Armstrong or Tori Amos? |
| WFMT FM radio transmits from the second tallest building in the United States, which is located where? |
| Who was the recipient of a prize also given to a player for Chinese club Tianjin Quanjian? |
| Which of Tara Strong major voice role in animated series is an American animated television series based on the DC Comics fictional superhero team, the "Teen Titans"? |

| **ComplexWebQuestions** |
|---|
| What university has more than 15,835 undergraduates and is the university Derek Fisher attended? |
| Who influenced Whitman's poetry who was the public speaker who spoke about the American Civil War? |
| What is the main train station called in the governmental jurisdiction where the government includes the position Mayor of San Francisco? |
| Which country that borders Russia has the smallest ISO? |
| What country that's a neighbor of Russia is a governmental jurisdiction where Erik Asanbayev holds a governmental office? |

| **QAMParI** |
|---|
| Where did a Roman Catholic archbishop of San Francisco attend school? |
| At what institution did a Bishop of Derby receive their education? |
| For which movie did Mani Ratnam work on the script and serve as producer? |
| What Type VII C/41 and Type VII ships was in both the vessel classes of German? |
| Philip Kaufman was responsible for both writing and directing of which movie? |

Table 1: Randomly sampled examples from RoMQA and other compositional QA datasets. Human evaluations show that people are more likely to ask RoMQA questions than those from other compositional QA datasets. Qualitatively, RoMQA questions exhibit fewer artifacts such as overly precise constraints (e.g. 15,835 undergraduates), overly numerous references (e.g. is an American animated... based on... the "Teen Titans"), and unusual attribute comparisons (e.g. smallest ISO).

sition frequency in the KB (Appendix A). This down-samples over-represented propositions such as `country`. We keep clusters with $\geq 3$ constraints to be able to generate many related questions from each cluster. We discard clusters of potentially spuriously related constraints with a single shared answer. 10k clusters are randomly chosen for training and 15k clusters for evaluation.

**Sampling constraints to form logical queries.** We generate $\leq 5$ **logical queries** using each cluster. For each logical query, we copy the cluster and remove constraints with probability 0.1 and negate with 0.1. We negate sparingly because large numbers of negative constraints result in unnatural questions. We further remove redundant constraints (e.g. American presidents born in the US), and uniformly subsample up to 4 constraints. This constitutes a logical query with multiple conjunctions and subtractions. For instance, the cluster {`occupation pianist subj, born_in Paris subj`} can form a logical query `occupation`

`pianist subj AND born_in Paris subj`. We discard overly general queries with $\geq 5000$ answers.

**Creating natural language questions.** Mechanical Turk workers annotate logical queries marked with Wikidata titles, descriptions, and aliases into **questions**. Appendix B Figure 11 shows the interface. Two more annotators verify each annotation to confirm that it matches the logical query. We keep only annotations with 100% agreement, resulting in 11% being discarded. After verification, we additionally discard clusters with $\leq 2$ questions.

### 2.2 Dataset analyses and comparison

We compare RoMQA to prior compositional QA datasets: HotpotQA (Yang et al., 2018), ComplexWebQuestions (CWQ; Talmor and Berant, 2018), and QAMParI (Amouyal et al., 2022).

**Dataset size and question complexity** Table 2 shows that only RoMQA evaluates robustness to input variations. Moreover, only RoMQA and QAMParI are human-written with multiple answers

| Dataset | Train | Dev | Test | Human written | Multi answer | Gold evidence | Robustness evaluation |
|---|---|---|---|---|---|---|---|
| RoMQA (Ours) | 11k | 7k | 11k | Yes | Yes | Yes | Yes |
| HotpotQA | 90k | 7k | 7k | Yes | No | Yes | No |
| CWQ | 28k | 3k | 3k | Yes | Yes | No | No |
| QAMParI | 64k | 1k | 1k | Eval only | Yes | Yes | No |

Table 2: Dataset size and question complexity.

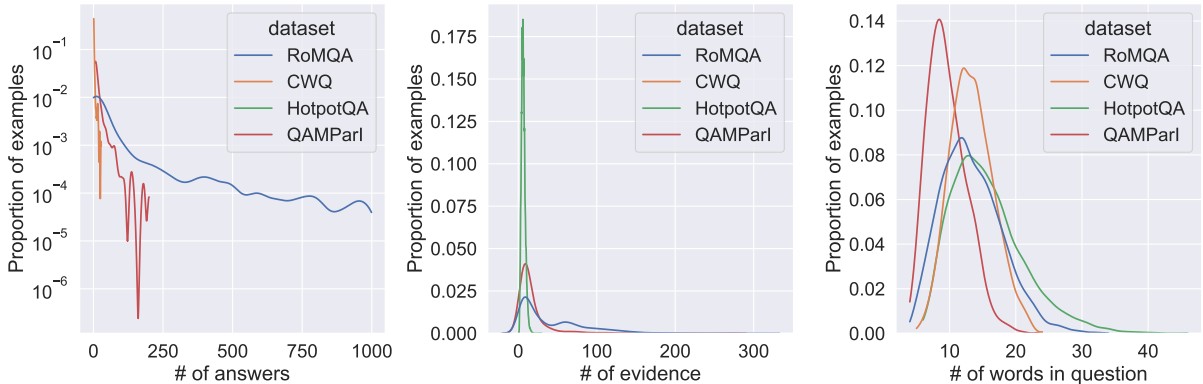

Figure 2: Dataset comparison over question, evidence, and answer size distributions.

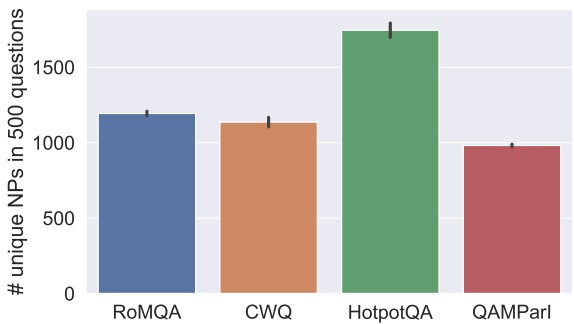

Figure 3: Question diversity as measured # unique noun-phrases in 500 randomly sampled questions from the development set of each dataset. The batches are randomly sampled 4 times to compute standard deviation.

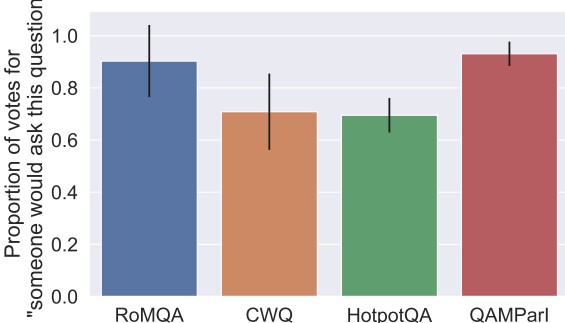

Figure 4: The distribution of questions naturalness ratings by 3 annotators on 1,000 randomly sampled questions from the development set of each dataset. Each annotator rates four questions shuffled in random order, one from each dataset. he annotator is asked whether they would ask the question themselves, and if they think someone else would ask the question.

and gold evidence. QAMParI provides 2k human-written evaluation questions while RoMQA provides 11k training and 17k evaluation questions.

Figure 2 shows the distribution of answer, evidence, and question sizes. First, RoMQA questions require finding many answers. On average, RoMQA questions have 108 answers — at least 10x larger than others. Second, RoMQA requires reasoning over a much more evidence documents. On average, entities in the RoMQA answer set combine for a total of 52 evidence sentences. Third, RoMQA questions are longer with more words. Figure 3 shows that, in a random sample of 500 questions, RoMQA refers to more unique noun phrases apart from HotpotQA.

**Naturalness human evaluation** Prior work sometimes sacrifice question naturalness in pursuit of complexity. Table 1 illustrates artifacts

in randomly sampled questions from HotpotQA, ComplexWebQuestions, and QAMParI. They include unusual constraints such as IDs (e.g. . . . has the smallest ISO?[1]), overly precise constraints (e.g. . . . is an American animated television series based on the DC Comics fictional superhero team, the "Teen Titans") and an excessive number of referential expressions (e.g. . . . in the governmental jurisdiction where the government includes the position Mayor of San Francisco). We compare the naturalness of 1,000 randomly sampled human written questions from each dataset. Each annotator is shown a randomly sampled question from each

[1]ISO codes are 2-3 character-long codes that represent names of countries and their subdivisions.

| Model class | Setting | Input format |
|---|---|---|
| Zero-shot | Closed | Gilbert Amy [SEP] Who was a pianist born in Paris? |
| | Open | Who was a pianist born in Paris? |
| Few-shot | Closed | Katie Bell [SEP] Who is an athlete who participated in diving, and was born in Stockholm? [SEP] False [newline] . . . Gilbert Amy [SEP] Who was a pianist born in Paris? [SEP] True |
| | Open | Who is an athlete who participated in diving, and was born in Stockholm? [SEP] Johan Jansson . . . [newline] . . . Who was a pianist born in Paris? [SEP] |
| Supervised | Closed | Gilbert Amy [SEP] Who was a pianist born in Paris? |
| | Open | Who was a pianist born in Paris? |
| Sup+evidence | Closed | Gilbert Amy [SEP] Who was a pianist born in Paris? [SEP] Gilbert Amy (born 29 August 1936) is a French composer and conductor . . . |
| | Open | Who was a pianist born in Paris? [SEP] Gilbert Amy (born 29 August 1936) is a French composer and conductor . . . |

Table 3: Input format for question "Who was a pianist born in Paris". For closed setting, "Gilbert Amy" is used as example candidate. Evidence includes retrieved sentences for retrieval methods or gold evidence for upperbound.

dataset, shuffled in random order. The annotator is asked "how likely would you ask this question?" and "how likely do you think another person would ask this question?" Each question is annotated by 3 crowdworkers. The breakdown of ratings across questions is shown for each dataset in Figure 4. On average, annotators consider RoMQA to be significantly more natural than HotpotQA and ComplexWebQuestions.

## 3 Experiments

How do existing QA systems perform on RoMQA? Are they robust to variations in question constraints? We answer these questions by experimenting with state-of-the-art models in zero-shot, few-shot, and supervised learning settings.

### 3.1 Evaluation

**Open vs closed settings** Given a question in RoMQA, a QA system should produce a set of answer entities. In the **closed setting**, the system is given 100 candidate entities and must identify which ones answer the question. Negative candidates are potentially difficult for a model because they can match any constraint in the question. In the **open setting**, candidates are not given.

**Evaluation metrics** RoMQA questions have many answers. For the closed setting, we evaluate predictions using $F_1$ and accuracy. $F_1$ measures set overlap between predicted and gold answers, while accuracy measures whether they match exactly. For the open setting, we evaluate precision@K ($P_{10}$) for two reasons. First, precision gives partial credit when it is too hard to enumerate the full set. Second, the user may ask a question with many answers (e.g. hundreds or thousands) with the intent

of only seeing some examples (e.g. list British footballers). For each score, we additionally have a **robustness variant**. Let $Q = \{q_1, q_2 \ldots q_n\}$ denote a cluster of $n$ related questions. The question $q_i$ has the corresponding predicted answer set $p_i$ and gold answer set $g_i$. A robustness score is the worst-case score across the cluster. For instance, the robust $F_1$ is $F_1{}^R(Q) = \min_i (F_1(p_i, g_i))$. We compute similar robustness scores for accuracy and precision@K.

### 3.2 Models

We evaluate three classes of models. The input format for each class is shown in Table 3.

**Zero-shot.** We consider a naive closed setting baseline that predicts all candidates as answers. We also include state-of-the-art prompting models tk-instruct-3B (Wang et al., 2022) and opt-instruct-175B (Zhang et al., 2022). In the closed setting, they generate yes/no given the question and a candidate. In the open setting, they generate answers given the question.

**Few-shot in-context learning.** We evaluate tk-instruct-3B (Wang et al., 2022), opt-instruct-175B (Zhang et al., 2022), and GPT3 (text-davinci-002; Brown et al. (2020)) with as many in-context examples as the model allows (4, 8, and 8 respectively). Input format is similar to the zero-shot setting, with the addition of in-context examples. In closed setting, the context includes an equal number of randomly sampled positive and negative examples. We compare the scores for the candidate answering vs. not answering the current question. These scores are calibrated using channel calibration (Min et al., 2022). In open setting, the model context includes

| Model | Dev | | | | Test | | | |
|---|---|---|---|---|---|---|---|---|
| | $F_1$ | $F_1{}^R$ | acc | $acc^R$ | $F_1$ | $F_1{}^R$ | acc | $acc^R$ |
| **Zero-shot** | | | | | | | | |
| predict all | 34.1 | 13.1 | 0.0 | 0.0 | 33.5 | 12.8 | 0.0 | 0.0 |
| tk-instruct-3b-0shot | 34.5 | 13.2 | 0.0 | 0.0 | 34.0 | 12.8 | 0.0 | 0.0 |
| opt-instruct-0shot | 36.0 | 14.0 | 0.0 | 0.0 | 36.0 | 14.0 | 0.0 | 0.0 |
| **Few-shot** | | | | | | | | |
| tk-instruct-3b-4shot | 33.5 | 12.9 | 0.0 | 0.0 | 33.1 | 12.5 | 0.0 | 0.0 |
| opt-8shot | 38.9 | 16.1 | 0.0 | 0.0 | 38.5 | 15.5 | 0.0 | 0.0 |
| **Supervised** | | | | | | | | |
| binary | 35.9±0.7 | 10.9±0.6 | 2.3±0.1 | 0.3±0.1 | 35.5±1.8 | 10.2±1.3 | 2.5±0.3 | 0.3±0.1 |
| binary+retrieval | 64.0±0.6 | 38.6±1.1 | 7.0±0.3 | 0.3±0.1 | 63.8±0.5 | 37.9±1.1 | 7.0±0.1 | 0.7±0.1 |
| binary+gold evidence | 95.3±0.3 | 83.5±0.8 | 72.3±1.8 | 39.2±2.6 | 95.0±0.3 | 83.4±0.9 | 71.5±0.9 | 38.2±1.2 |

Table 4: Model performance on closed setting RoMQA. Metrics are set $F_1$, set accuracy, and their robustness counterparts (i.e. worst case measure over cluster of related questions). Each model is given 100 candidate entities and must decide which entity belongs to the answer set. The retrieval model additionally observes sentences retrieved via BM25 followed by DPR. Zero-shot and few-shot are binary-classifiers calibrated with channel calibration. Supervised models fine-tune `BART-large` on the training data to classify the answer set on a per-entity basis.

≤10 subsampled answers for each example.

**Supervised learning.** We tune `BART-large` with and without retrieved evidence (Lewis et al., 2020) and show standard deviation across 5 random seeds. For the closed setting which considers a candidate entity, we use a two-stage hybrid retrieval because dense retrievers under-perform sparse retrievers on rare, precise entities (Sciavolino et al., 2021). We first retrieve documents with BM25 using entity title as the query. We then use DPR (Karpukhin et al., 2020) to retrieve document sentences whose cosine similarity with the question exceed a threshold (0.65). This threshold is tuned by maximizing the retrieval hit rate on the validation set. Finally, we fine-tune to classify whether each candidate belongs to the answer set.

In the open setting, we do not use classification models because it is computationally prohibitive to decide over all possible (2.9M) entities per question.[2] Instead, we directly retrieve evidence with DPR and fine-tune the model to generate the answer set as a 1024-token sequence.

**Upper bound with gold evidence.** We provide a performance upper bound by training supervised models with gold evidence — sentences that provide provenance to an implicit question constraint. For instance, consider the example in Figure 1. *Claude Hellfer* is an answer to the question "Who was a pianist born in Paris?", whereas *David Fray* is not. In this case, the gold evidence includes

---

[2]For reference, inference over the entire test set with 100 candidate entities per example using the classification model requires 10 hours on a Volta 32GB GPU.

"*Claude Helffer… was a French pianist*", "*Helffer was born in Paris*", and "*David Fray… is a French classical pianist*". Because the gold evidence only contains sentences that provide provenance to an implicit constraint, it does not contain the sentence "*David Fray was born in Tarbes, near the Pyrenees.*" In other words, given gold evidence, the QA model does not need to filter out incorrect answers in the candidate answers pool (e.g. *David Fray*) because it only has to verify that the provided evidence supports all parts of the question (e.g. born in Tarbes instead of Paris). Instead, it only needs to verify that the evidence references all implicit constraints. Consequently, the gold evidence setting is overly optimistic in that part of the reasoning is completed by a perfect retriever. While no such retriever currently exists, this setting nevertheless provides an upper bound estimate for RoMQA.

### 3.3 Results

Table 4 and Table 5 show performance on RoMQA closed and open settings. RoMQA is challenging for state-of-the-art large-scale models. Moreover, these models are not robust to variations in question constraints. The best models significantly trail the gold evidence upper bound, showing there is significant room future work.

**Zero-shot and few-shot models perform similarly to naive predict-all baseline.** In the closed setting, each system is given a set of 100 candidate entities and must identify which entity belong to the answer set. We find that state-of-the-art pre-

| Model | Dev | | | | Test | | | |
|---|---|---|---|---|---|---|---|---|
| | $F_1$ | $F_1{}^R$ | $P_{10}$ | $P_{10}{}^R$ | $F_1$ | $F_1{}^R$ | $P_{10}$ | $P_{10}{}^R$ |
| **Zero-shot** | | | | | | | | |
| tk-instruct-3b-0shot | 0.2 | 0.0 | 1.7 | 0.0 | 0.3 | 0.0 | 1.8 | 0.0 |
| opt-instruct-0shot | 1.6 | 0.0 | 5.0 | 0.1 | 1.6 | 0.1 | 5.1 | 0.2 |
| **Few-shot** | | | | | | | | |
| tk-instruct-3b-4shot | 0.3 | 0.0 | 1.9 | 0.0 | 0.4 | 0.0 | 2.0 | 0.0 |
| opt-8shot | 2.1 | 0.0 | 5.5 | 0.2 | 2.2 | 0.1 | 5.6 | 0.1 |
| gpt3-8shot | 4.3 | 0.4 | 13.3 | 1.2 | 4.4 | 0.4 | 12.6 | 1.0 |
| **Supervised** | | | | | | | | |
| seq2seq | 32.2±0.9 | 11.2±0.5 | 47.3±1.2 | 20.3±1.2 | 32.6±1.0 | 11.5±0.7 | 45.7±0.8 | 19.7±1.1 |
| seq2seq+retrieval | 41.6±0.8 | 19.1±0.7 | 58.6±1.5 | 38.3±2.4 | 41.0±0.5 | 18.4±0.6 | 56.8±1.4 | 36.0±2.3 |

Table 5: Model performance on open setting RoMQA. Metrics are set $F_1$, precision at 10, and their robustness counterparts (i.e. worst case measure over cluster of logically related questions). Each model is given the question and must generate the answer set as a sequence. The retrieval model additionally observes sentences retrieved by DPR. Supervised models fine-tune `BART-large` on the training data. All models generate the answer set as a sequence. We do not evaluate upperbound gold evidence method because it necessarily provides candidate entities and therefore is no longer open domain.

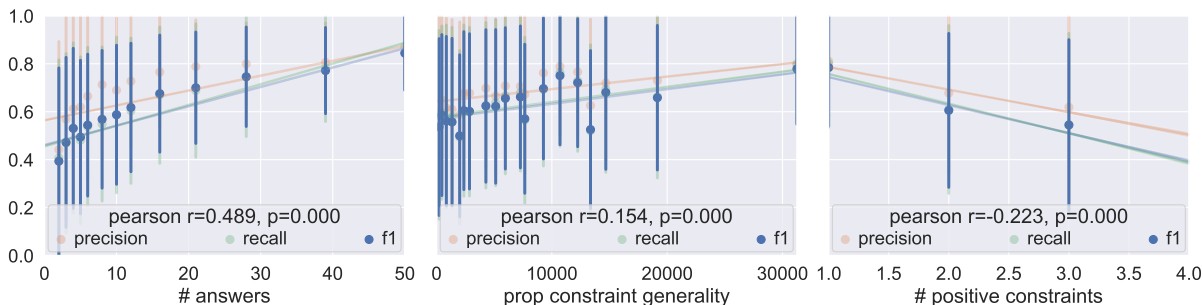

Figure 5: Correlation with model performance ($F_1$) on the closed setting. Imprecise questions with many answers are easier to answer (higher $F_1$). Questions based on general propositions that co-occur with many different entities are easier to answer. Questions with more constraints are more difficult to answer.

trained instruction prompting models perform on par with the naive baseline of simply predicting that every candidate belongs to the answer set. This occurs both with instructing prompting and in-context learning models, and suggests that they can not effectively reason about the types of compositional questions found in RoMQA.

**Both closed and open settings remain challenging with supervised training.** When given 11k annotated examples, large retrieval models perform better than zero-shot and few-shot LMs. However, supervised performance also trails the gold evidence upper bound. This suggests that there is significant room for modeling improvements that retrieve and compose evidence in order to answer compositional questions.

What types of questions do the best-performing supervised systems struggle with? Figure 5 plots Pearson correlation with $F_1$ in the closed setting, and shows that systems generally struggle with more precise questions.[3] First, when the question has many answers, the model has an easier time producing some correct answers. Second, the model performs better on more general propositions that co-occur with many different unique entities. Third, the model struggles with questions with more implicit constraints.

**Methods are not robust to question constraint variations.** All methods drop in performance when considering the worst-case performance among clusters of related questions. This suggests that large LM-based QA systems are not robust to variations in question constraints. Figure 6 shows what types of questions result in robustness drops. Compared to other questions in the same cluster, a question is easier if it contains more answers, and harder if it specifies more implicit constraints.

Training on clusters of related questions

---

[3]Pearson correlation is a measure of linear correlation. A Pearson coefficient of 1 or -1 implies positive or negative linear correlation, while 0 implies no linear dependency.

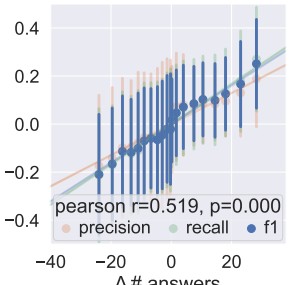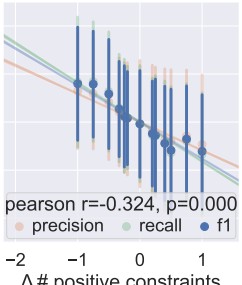

Figure 6: Correlation with robustness drop ($F_1$ - cluster mean $F_1$) in closed setting. The axes denote deviation from the cluster means. Among a cluster of related questions, a more precise question with more constraints or fewer answers tend to be harder for the model than related questions with more answers or less constraints.

(e.g. RoMQA clusters) is one way to improve model robustness. Given clusters of questions with related implicit constraints, in the first setting we train on unrelated questions — one question from each cluster for a total of K training examples. In the second setting, we train on related questions — K consecutive examples from entire clusters. Table 6 shows that while the diversity from training on unrelated questions marginally improves overall performance, training on clusters of related questions results in more robust models. Nevertheless, the robustness drops remain significant. Considering that variations in RoMQA questions are reasonable questions humans would write, as opposed to artificially created adversarial questions, our findings suggests that there is a practical need for developing more robust QA systems.

**Building context for open setting is very challenging.** While the closed setting RoMQA challenges current state-of-the-art models, the open setting remains an even greater challenge. A key challenge in the open setting is that it is difficult to compute the evidence set required the answer the question. Consider Figure 1's question "Who was a Western classical music pianist, not born in Paris". The obvious way a human would answer this question is substracting the set of people born in Paris from the set of Western classical music pianists. However, both of these sets are very large. Our results show that an end-to-end large language model struggles in reasoning over such large sets.

## 4 Related Work

**Question answering datasets** Existing QA datasets focus on answering from a single passage (Rajpurkar et al., 2016; Joshi et al., 2017;

| Training questions | Closed $F_1$ | $F_1^R$ | $\Delta F_1$ | Open $P_{10}$ | $P_{10}^R$ | $\Delta P_{10}$ |
|---|---|---|---|---|---|---|
| Unrelated | 56.2 | 28.2 | -28.0 | 27.0 | 1.6 | -25.4 |
| Related | 55.7 | 28.6 | -27.1 | 26.3 | 11.7 | -14.6 |

Table 6: Training supervised retrieval models on related vs. unrelated questions. Unrelated questions training involves selecting one question from each cluster, . In contrast, related questions training includes entire clusters of related questions. While training on a more diverse set of unrelated questions yields slightly higher overall performance, training on related questions produces more robust models.

Kwiatkowski et al., 2019; Sciavolino et al., 2021) to answering over multiple passages (Yang et al., 2018; Welbl et al., 2018; Thorne et al., 2018). Recent work emphasize answering questions that have multiple answers (Min et al., 2020; Amouyal et al., 2022). RoMQA combines the latter two settings in that it requires answering questions over multiple pieces of evidence to provide multiple answers. Compared to prior datasets, RoMQA questions require robust reasoning over more pieces of evidence to provide more answers.

**Robustness evaluation** NLP systems have previous been show to lack robustness. They are susceptible to character based attacks that comprise of both nonsensical inputs (Jia and Liang, 2017), random sentence/word triggers (Jia and Liang, 2017; Wallace et al., 2019), and semantically equivalent inputs adversarially selected to disrupt system behaviour (Ribeiro et al., 2018; Zhao et al., 2018; Iyyer et al., 2018). In contrast, the questions in RoMQA are not adversarial — they are written with reasonable information-seeking intent.

**Zero-shot/few-shot learning** Recent work has also shown that large pretrained LMs can perform zero-shot and few-shot inference (Brown et al., 2020; Wang et al., 2022; Zhang et al., 2022). For the former, the LM performs inference given a prompt or an instruction. For the latter, the LM is also given training examples as demonstration. We use both settings as baselines for RoMQA, and find that there is significant room for improvement in large-scale pretraining to capture compositional reasoning over multiple pieces of evidence text.

## 5 Conclusion

We presented RoMQA, the first benchmark for robust, multi-evidence, multi-answer QA. RoMQA evaluates robustness of models to varying question constraints by testing for worst-case

performance among clusters of related questions. RoMQA is a challenging dataset for state-of-the-art large LMs and provides a quantifiable test for developing more robust QA methods.

# 6 Limitations

**Robustness definition** A robust system needs to be robust towards all forms of input variations. The type of robustness to varying implicit constraints measured in RoMQA is but one form. Other forms of robustness include adversarial character attacks, sentence injections, and word triggers. RoMQA is limited to English — future work should investigate robustness in other and multiple languages. The questions in RoMQA are based on Wikidata triples and contains limitations inherent to Wikidata and large-scale knowledge-graphs in general (e.g. staleness of facts, over-represented propositions/entities, incompleteness). The models evaluated in this work (e.g. `opt-instruct-175B`, `text-davicinci-002`) are but a sample of state-of-the-art language models at the time of writing. As new models emerge, it is possible that they exhibit robustness beyond that of the evaluated models.

# 7 Ethics Statement

In order for AI systems to be used in critical applications, we need to establish trust in these systems. One hallmark of trustworthiness is robust system behaviour. RoMQA is a preliminary step in quantifying the robustness of question answering systems to natural variations in the question. It is our hope that the release of RoMQA will facilitate the development of more robust and more trustworthy NLP systems, and that future work will provide further quantifiable tests for the robustness of NLP systems.

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

## A    Subsampling propositions

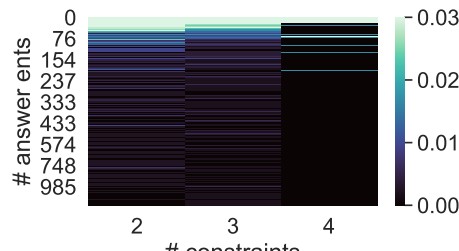

Figure 7: Number of constraints per cluster.

We want questions that cover diverse topics, however Wikidata has a very skewed proposition distribution, with a long tail of rare propositions. Hence, we down-sample frequent propositions. Let $P_{\text{prop}}(x)$ denote the percentage of triples that contain the proposition $x$. We define the average proposition probability as $P_{\text{prop}}' = \frac{1}{|X|} \sum_x P_{\text{prop}}(x)$. Given a constraint with proposition $x$, we remove it with probability $r = 1 - \min\left(1, \frac{P_{\text{prop}}'}{P_{\text{prop}}(x)}\right)^{\frac{1}{2}}$. In particular, those with below average frequency are not removed, and those with above average frequency are removed with increasing likelihood. After removing constraints based on propositions, we randomly sample up to 10 constraints using a distribution over their inverse proposition probabilities $\frac{1}{P_{\text{prop}}}$. Figure 7 shows the distribution over cluster sizes after resampling. Figure 10 shows that resampling results in a more diverse set of questions by emphasizing rarer propositions in the knowledge graph.

## B    Annotation

RoMQA questions are annotated by crowd-workers on Amazon Mechanical Turk from the US, Canada, UK, Australia, and New Zealand. We require that annotators have $\geq 95\%$ approval rating and have done a minimum of 5000 HITs. Questions are submitted for annotation in batches of 500. For each batch, a sample of 10 questions from each worker is inspected by the authors. If $\geq 2$ of annotations in the sample are incorrect, then response from the worker in that batch are marked for re-annotation. The final set of annotations are additionally verified by 2 more crowd-workers to confirm that they correspond to constraints. We keep only examples with 100% agreement, which corresponds to 89% of the annotated data. Workers were paid a minimum of 15 USD per hour, estimated using completion time per HIT.

## C    Dataset Statistics

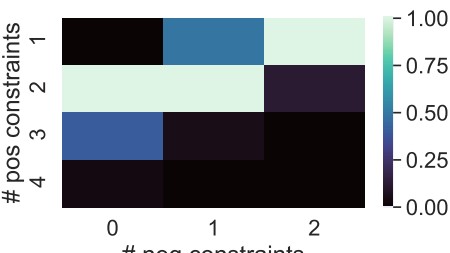

Figure 8: Answer set size vs constraint count.

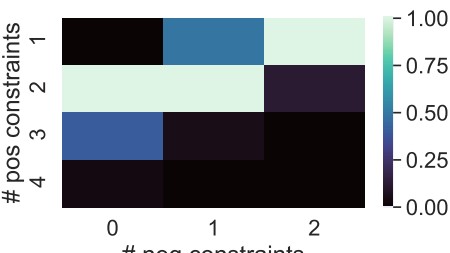

Figure 9: Positive vs negative constraint count.

**Cluster sizes.**    Figure 7 shows the distribution of cluster sizes in RoMQA. During the sampling procedure, we remove small clusters of $\leq 3$ questions and avoid large clusters of $\geq 7$ questions.

**Implicit constraint distribution.**    Figure 9 shows the distribution of positive and negative implicit constraints in RoMQA questions. Most questions have 2 positive constraints, and 0-1 negative constraints. We limit questions to 7 constraints during sampling. In practice, nonsensical questions with too many constraints are filtered out during verification. Figure 8 shows the distribution of implicit constraints vs. the size of the answer set. More precise questions with more implicit constraints typically have fewer answers. In general, RoMQA questions may have more than 1000 answers, though the vast majority contain less than 1000 answers. The outlier questions with more than 1000 answers are not shown in the figure.

## D    Experiment setup

For zero-shot and few-shot models, we use API services provided by the original authors. For supervised models, we fine-tune `BART-large` models with learning rate $1e-6$ on V100 GPUs. For classification in the closed setting, we train with batch size 100 and evaluate with batch size 1000. For generation in the open setting, we train and evaluate with batch size 2 and decode with beam size 3. Experiments are run on a Slurm cluster of V100

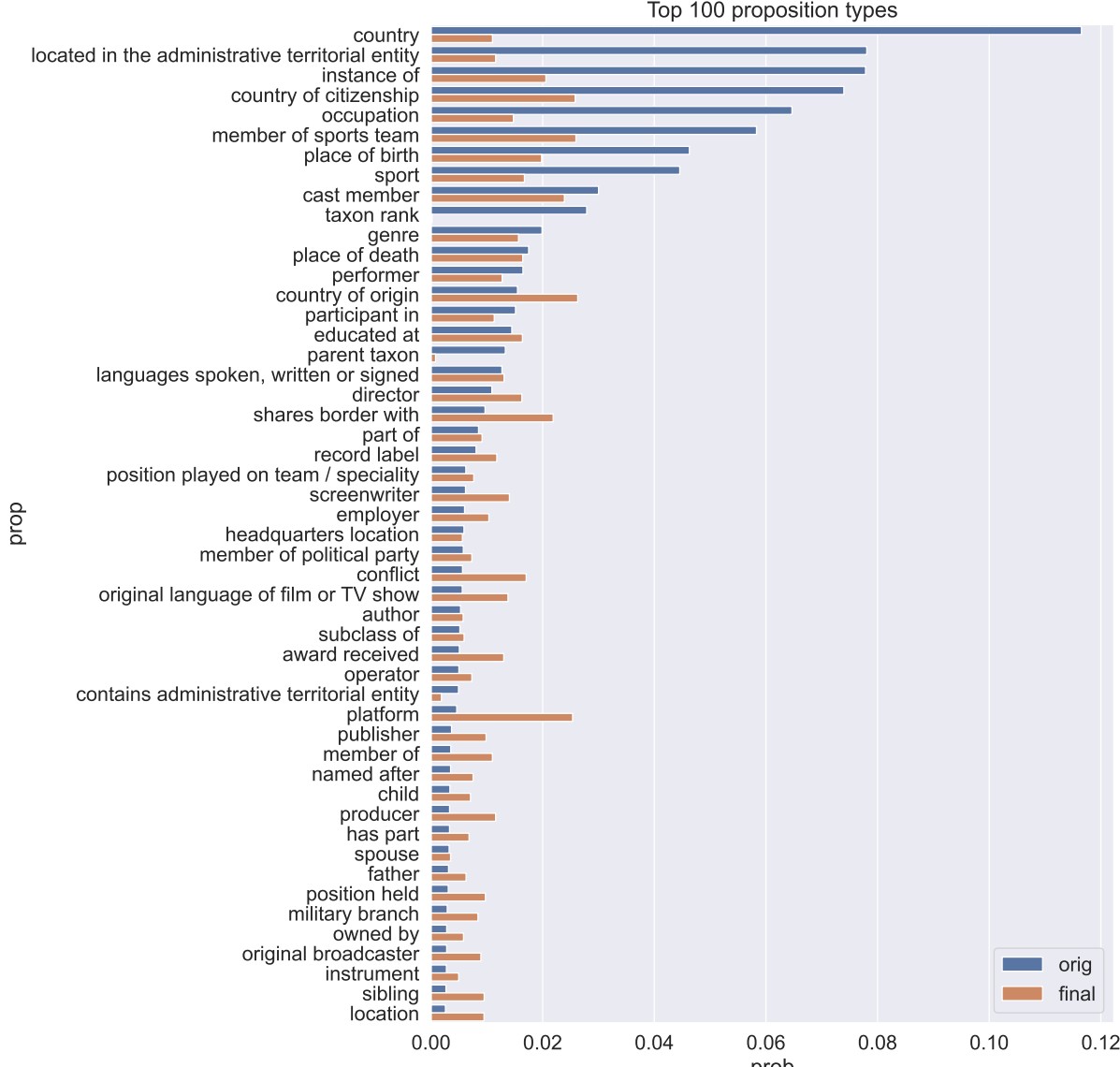

Figure 10: Most common propositions, before and after subsampling. Subsampling downsamples overly represented propositions in the knowledge graph and results in a more diverse set of propositions and question topics.

GPUs for 3-5 days each. We use default hyperparameters for supervised models, with maximum batch size as allowed by our GPUs. We perform early stopping on the validation set (for supervised methods) and evaluate on the test set only once.

# E License

The majority of RoMQA is licensed under CC-BY-NC, however portions of the project are available under separate license terms:

- qwickidata: Apache 2.0

- hydra-core: MIT

- torch: link

- tqdm: MIT

- rank_bm25: Apache 2.0

- spacy: MIT

- sentence_transformers: Apache 2.0

- ray: Apache 2.0

- wrangl: Apache 2.0

- ujson: link

## Write a paragraph summarizing these statements:

| | | | |
|---|---|---|---|
| True | Damon Hill: British racing driver | pole position: person, who starts race at first row | this thing |
| False | 1993 Formula One season: sports season | part of: object of which the subject is a part. AKA: in, chain, contained within | this thing |

Examples of things that should meet the criteria described by your paragraph (don't copy them in your answer):

- 1996 Argentine Grand Prix: Formula One motor race held in 1996
- 1995 Australian Grand Prix: 581st Formula 1 Championship Grand Prix
- 1994 French Grand Prix: Formula One motor race held in 1994
- ...

Figure 11: Mechanical Turk annotation interface for question writing. The annotator is shown a collection of positive and negative constraints in random order. Each constraints consists of an entity, a proposition, and a direction. Entities and propositions are expanded with descriptions and aliases from Wikidata. A subset of answer entities is listed to disambiguate answer types.