# OpenReview forum: "RoMQA: A Benchmark for Robust, Multi-evidence, Multi-answer Question Answering"
_EMNLP/2023/Conference — EMNLP 2023 Findings_

### Official Review · Reviewer_zaFs · 2023-08-01

**Typos Grammar Style And Presentation Improvements:** l. 111, such "as" RoMQA clusters
**Soundness:** 4

**Excitement:**

3: Ambivalent: It has merits (e.g., it reports state-of-the-art results, the idea is nice), but there are key weaknesses (e.g., it describes incremental work), and it can significantly benefit from another round of revision. However, I won't object to accepting it if my co-reviewers champion it.

**Missing References:**

2. Multi-evidence retrieving is a popular topic in NLP, it would be benefit if the author adds discussions and comparisons of multi-evidence retrieving in the paper.

**Paper Topic And Main Contributions:**

This paper presents a new QA dataset called RoMQA. RoMQA includes clusters of similar questions that share certain implicit constrains, require multi-evidences retrieval, and have multiple answers. The dataset is evaluated with 3 experiment settings: zero-shot, few-shot model, and supervised model, all with closed and open input format.
Results show that existing QA systems are not robust to variation within a cluster of questions.

**Questions For The Authors:**

1. In Table 1 under RoMQA, why there are two sentences that are not questions, are they included in the dataset?
2. l. 243, in the closed setting, how are the 100 candidates selected? Are they different for different questions?

**Reasons To Accept:**

1. The paper present a new dataset with high quality, which can be considered as a main contribution.
2. Creating clusters of similar questions based on shared constraints seems effective.
3. The paper is well-written and easy to follow. Experiments are thorough and solid.


**Reasons To Reject:**

1. The author defines robustness of the model as its capability to a cluster of questions that share multiple constraints. However, the robustness in general is considered broader than this.
2. The results show that the model is not robust to a cluster of similar questions, it would be interesting to see how the shared constraints (or contradicted constraints) impact the model performance.


**Reproducibility:**

5: Could easily reproduce the results.

**Reviewer Confidence:**

4: Quite sure. I tried to check the important points carefully. It's unlikely, though conceivable, that I missed something that should affect my ratings.

---

> ### Author Rebuttal · Authors · 2023-08-28
>
> Thanks for your insightful comments! Please see our responses below.
>
> > ***1. The author defines robustness of the model as its capability to a cluster of questions that share multiple constraints. However, the robustness in general is considered broader than this.***
>
> We agree that the definition of robustness is considered broader and contains different aspects. They are all important. We choose to focus on robustness to compositional constraints because a reliable compositional QA system should exhibit the ability to manage minor variations in constraints, demonstrating a deep understanding of the text and the relationships between different pieces of information.
> We believe it would be valuable for future study to broaden the scope to include other dimensions of robustness. We appreciate the reviewer for highlighting this perspective and will include it into our discussion of potential avenues for future research. We will also clarify the scope of this work to compositional constraints.
>
>
> > ***2. The results show that the model is not robust to a cluster of similar questions, it would be interesting to see how the shared constraints (or contradicted constraints) impact the model performance.***
>
> As shown in Figure 6, we show that very specific questions, with a lot of constraints or limited possible answers, are usually harder for the model to handle compared to questions that have more possible answers or fewer constraints. For instance, typically, given that the answer set is not very large (or considering the closed setting), the question “What are the A that B and C” is more difficult to answer than “What are the A that B”. We do not evaluate for contradicting constraints that result in no answers, but will investigate this in the future.
>
>
> > ***3. In Table 1 under RoMQA, why there are two sentences that are not questions, are they included in the dataset?***
>
> We use the term “question” to refer to utterances that have information-seeking intent. We will clarify this in the manuscript. Thank you for pointing this out.
>
> > ***4. l. 243, in the closed setting, how are the 100 candidates selected? Are they different for different questions?***
>
> Thank you for this question - we will add details on how the 100 candidates are generated in the paper. Given a set A of positive entities (e.g. those that answer the question), we sample at most 50 candidates. Then, we consider all entities that answer one of the constraints in the question (call this B), and sample the remaining entities from the set B-A. In this way, the negative candidates are “hard” because they are very related to the question in the sense that they answer at least one of the constraints.
>
> > ***5. Multi-evidence retrieving is a popular topic in NLP, it would be beneficial if the author adds discussions and comparisons of multi-evidence retrieving in the paper.***
>
> We thank the reviewer for pointing us to the related works. IIn the final version of our paper, we will include a discussion on multi-evidence retrieval and incorporating the following papers
>
> [1] Answering Complex Open-Domain Questions with Multi-Hop Dense Retrieval
>
> [2] Multi-hop Evidence Retrieval for Cross-document Relation Extraction
>
> [3] Multi-Hop Paragraph Retrieval for Open-Domain Question Answering
>
> If there are any additional related works that we may have overlooked, we would appreciate your guidance in identifying them so that we can include them in our final version.

---

### Official Review · Reviewer_EMCn · 2023-08-06

**Soundness:** 2

**Excitement:**

3: Ambivalent: It has merits (e.g., it reports state-of-the-art results, the idea is nice), but there are key weaknesses (e.g., it describes incremental work), and it can significantly benefit from another round of revision. However, I won't object to accepting it if my co-reviewers champion it.

**Paper Topic And Main Contributions:**

The paper proposes a new multi-evidence, multi-answer question answering (QA) dataset that was generated by mining entities and constraints from wikidata and having annotators convert the logical form of these entities and constraints to questions. The authors cluster related questions together and use a 'robustness' score by only considering the minimum performance of the questions in the same cluster. In addition, they explore the performance of zero-shot, few-shot (in-context learning) and fine-tuning of language models on the dataset and show that the dataset is challenging. While the dataset is useful, the modeling part of the paper needs further details in order to support the claims made in the paper.

**Questions For The Authors:**

- Line305: how was the 0.65 threshold determined
- Line 300: It would still be interesting to evaluate the performance of a dense retriever.
- Line 314: The section on "upper bound with gold evidence" is unclear.

**Reasons To Accept:**

- The authors propose a large dataset for question answering for entity-based questions which will be useful for other researchers.

**Reasons To Reject:**

- The modeling part of the paper where the authors mention that the performance of prompting LLMs is on par with the naive baseline is not convincing.
- Some parts of the write-up is unclear and some details are missing (please see questions below)

**Reproducibility:**

3: Could reproduce the results with some difficulty. The settings of parameters are underspecified or subjectively determined; the training/evaluation data are not widely available.

**Reviewer Confidence:**

1: Not my area, or paper was hard for me to understand. My evaluation is just an educated guess.

---

> ### Author Rebuttal · Authors · 2023-08-28
>
> Thanks for your insightful comments! Please see our responses below.
>
> >***1. The modeling part of the paper where the authors mention that the performance of prompting LLMs is on par with the naive baseline is not convincing.***
>
> We ask the reviewer whether our finding is “not convincing” or “surprising”. If the reviewer finds our result not convincing, we welcome feedback on how to make our finding more convincing. If the reviewer finds our result surprising: our dataset comprises complex questions, each with multiple possible answers—averaging 108 answers per question. However, large-scale language models typically generate only a handful of answers, which inherently restricts their performance on our dataset. Moreover, the observed suboptimal performance of current state-of-the-art models highlights the complexity of the challenges posed by our dataset, thereby emphasizing the need for future research in the direction of compositional QA. We hope that this addresses the reviewer’s concerns.
>
> >***2. Line305: how was the 0.65 threshold determined.***
>
> This threshold is tuned by maximizing the retrieval hit rate on the validation set
>
> >***3. Line 300: It would still be interesting to evaluate the performance of a dense retriever.***
>
> Our early experiments showed that dense retrieval underperforms hybrid retrieval on a subset of the corpus. This finding is in line with other works on sparse retrieval.
>
> >***4. Line 314: The section on "upper bound with gold evidence" is unclear.***
>
> In that section, we aim to emphasize that gold evidence only includes sentences that support the answer, and not sentences that would contradict it or are irrelevant. This means that, when using gold evidence, the model does not need to filter out incorrect answers in the candidate answers pool because it only has to verify that the provided evidence supports all parts of the question. We appreciate your feedback in terms of writing and will clarify this section in the final version of the paper.

---

### Official Review · Reviewer_Tpkz · 2023-08-09

**Soundness:** 3

**Excitement:**

3: Ambivalent: It has merits (e.g., it reports state-of-the-art results, the idea is nice), but there are key weaknesses (e.g., it describes incremental work), and it can significantly benefit from another round of revision. However, I won't object to accepting it if my co-reviewers champion it.

**Paper Topic And Main Contributions:**

This paper introduces RoMQA, a new question answering dataset and benchmark for evaluating the robustness of QA systems. The paper claims that prior QA datasets do not explicitly test for model robustness to small variations in question phrasing or constraints and the authors aim to create a benchmark that measures QA robustness.

The main contributions are
1. The proposed RoMQA dataset containing clusters of related questions that vary in their implicit constraints which aim to address the robustness evaluation for the QA models.
2. The proposed RoMQA has more complex questions requiring reasoning over more evidence documents and providing more answers on average and the human evaluations show RoMQA questions are more natural than other compositional QA datasets.
3. Experiments show state-of-the-art models have low robustness based on RoMQA evaluation and the analysis indicates some interesting results e.g. training on RoMQA clusters improves robustness compared to unrelated questions.

In summary, the paper introduces a new robustness-focused QA dataset and benchmark, and uses it to demonstrate that existing models lack robustness to subtle question variations. The dataset and robustness metric are the main contributions.

**Questions For The Authors:**

1. When construct the question, how to consider the robustness evaluated examples and the adversarial examples (I suppose there are no consideration of the adversarial examples now), and what will happen if we add the adversarial examples when evaluate the robustness (from both the problem definition aspect and the evaluation aspect)?

2. What are the performance of the non-LLM methods, in terms of the widely use performance metric (e.g. F1, EM) and the robustness metric?

3. Since the datasets have already provided the evidences, what is the retrieval role of the retrieval model (in Table 5) and why we need retrieval here and how it can improve the QA results?

**Reasons To Accept:**

1. Addresses an important issue that the robustness is a critical but under-studied area for QA systems. The proposed dataset provides a concrete way to measure it.
2. The paper thoroughly evaluates many state-of-the-art models and sets challenging benchmarks and provides useful insights into what types of questions are more challenging in terms of robustness.
3. It seems that the RoMQA dataset thoughtfully constructed based on sourcing constraints from a knowledge graph.

**Reasons To Reject:**

1. The construction of the datasets could contain some risks. For example, the scope is limited to a specific type of robustness based on constraint variations and does not address other aspects like adversarial examples. Besides, the source highly relies exclusively on Wikipedia/Wikidata for sourcing facts which could limit diversity and lead to factual errors.

2. The evaluation is mostly focused on large pretrained language models which does not cover a wide range of QA methods and support the initial statement of the exist QA models cannot achieve better robustness ability. Besides, the problem definition could be more clear.

3. Lacks detailed error analysis explaining why models struggle on certain questions and the robustness metric may not correspond well to real-world robustness if question variations are not natural or representative. More qualitative examples can be added for both illustrating the dataset construction and evaluation.

**Reproducibility:**

3: Could reproduce the results with some difficulty. The settings of parameters are underspecified or subjectively determined; the training/evaluation data are not widely available.

**Reviewer Confidence:**

3: Pretty sure, but there's a chance I missed something. Although I have a good feel for this area in general, I did not carefully check the paper's details, e.g., the math, experimental design, or novelty.

---

> ### Author Rebuttal · Authors · 2023-08-28
>
> Thanks for your insightful comments! Please see our responses below.
>
> > ***1. The construction of the datasets could contain some risks. For example, the scope is limited to a specific type of robustness based on constraint variations and does not address other aspects like adversarial examples.***
>
> We agree that the definition of robustness is considered broader and contains different aspects. They are all important. We choose to focus on robustness to compositional constraints because a reliable compositional QA system should exhibit the ability to manage minor variations in constraints, demonstrating a deep understanding of the text and the relationships between different pieces of information.
> We believe it would be valuable for future study to broaden the scope to include other dimensions of robustness. We appreciate the reviewer for highlighting this perspective and will include it into our discussion of potential avenues for future research. We will also clarify the scope of this work to compositional constraints.
>
> > ***2. Besides, the source highly relies exclusively on Wikipedia/Wikidata for sourcing facts which could limit diversity and lead to factual errors.***
>
> As stated on the Wikidata page [1], the content in Wikidata is verifiable, supported by authoritative sources of information. References are incorporated into item pages, serving as pointers to specific sources that substantiate the data presented in a statement. While Wikidata may contain imperfections, we believe it to be a valuable source of factual knowledge for assessing compositional robustness, given its extensive coverage of relational facts. Moreover,  Additionally, it is worth noting that numerous previous studies have established their QA benchmarks based on Wikidata [2,3], indicating it is widely recognized in the research community. We appreciate your concern and will incorporate a discussion of it in the limitations section of our paper.
>
>
> > ***3. The evaluation is mostly focused on large pretrained language models which does not cover a wide range of QA methods and support the initial statement of the existing QA models cannot achieve better robustness ability. Besides, the problem definition could be more clear. What are the performance of the non-LLM methods, in terms of the widely use performance metric (e.g. F1, EM) and the robustness metric?***
>
> We appreciate the reviewer's comment on the scope of our evaluation. To the best of our knowledge, the current state-of-the-art QA systems, such as FiD [4] and Atlas [5], are predominantly based on language models. If the reviewer has any suggestions regarding additional systems to evaluate, we would appreciate the suggestion and incorporate the evaluation of these systems into our final version.
>
> > ***4. When construct the question, how to consider the robustness evaluated examples and the adversarial examples***
>
> We appreciate the reviewer bringing up the important aspect of adversarial examples. Our focus in this study was on evaluating the model's robustness to variations in constraints, which is one important dimension of robustness in compositional QA. We believe that adversarial examples and constraint variations are complementary dimensions of robustness that collectively provide a comprehensive evaluation of a model's capabilities. For the evaluation perspective, we will include two metrics: one is to evaluate robustness towards constraint variations and the other is to evaluate models’ ability to handle adversarial questions.
>
> > ***5. Since the datasets have already provided the evidences, what is the retrieval role of the retrieval model (in Table 5) and why we need retrieval here and how it can improve the QA results***
>
> In practical scenarios, the evidence required for QA systems is not always readily available, requiring the QA systems to learn two key skills: (1) the retrieval of multiple pieces of evidence, and (2) reasoning across different pieces of information to construct an answer. Our open-book setting is designed to mimic this real-world process of answering compositional questions. Additionally, there are many prior work that focuses on multi-evidence retrieval for multi-hop questions, as illustrated by the following studies [6, 7, 8]
>
>
> [1] https://www.wikidata.org/wiki/Wikidata:Verifiability
>
> [2] QAMPARI: An Open-domain Question Answering Benchmark for Questions with Many Answers from Multiple Paragraphs
>
> [3] Beyond I.I.D.: Three Levels of Generalization for Question Answering on Knowledge Bases
>
> [4] Leveraging Passage Retrieval with Generative Models for Open Domain Question Answering
>
> [5] Atlas: Few-shot Learning with Retrieval Augmented Language Models
>
> [6] 'Answering Complex Open-Domain Questions with Multi-Hop Dense Retrieval'
>
> [7] 'Multi-hop Evidence Retrieval for Cross-document Relation Extraction'
>
> [8] 'Multi-Hop Paragraph Retrieval for Open-Domain Question Answering'

---

### Meta-Review · Area_Chair_QtUp · 2023-09-16

**Recommendation:** 3

**Metareview:**

The authors propose a dataset for QA where questions have many answers. The authors focus the evaluation on robustness where clusters of similar questions are considered.

The reviewers like the robustness focus and the way the dataset is constructed.

On the other hand some reviewers mention that the baselines might be weak, although in my mind using LLMs and other smaller capacity supervised models from the literature makes sense. Reviewers also mention that the paper does not deal with all aspects of robustness, and to that authors replied that the paper focuses on compositionally. Finally, reviewers ask for an error analysis and several specific clarifications that were raised as questions.

---

### Decision · Program_Chairs · 2023-10-07

**Decision:**

Accept-Findings

**Comment:**

The authors propose a dataset for QA where questions have many answers. The authors focus the evaluation on robustness where clusters of similar questions are considered.

The reviewers like the robustness focus and the way the dataset is constructed.

On the other hand some reviewers mention that the baselines might be weak, although in my mind using LLMs and other smaller capacity supervised models from the literature makes sense. Reviewers also mention that the paper does not deal with all aspects of robustness, and to that authors replied that the paper focuses on compositionally. Finally, reviewers ask for an error analysis and several specific clarifications that were raised as questions.